# When Students Patronize Fast-Food Restaurants near School: The Effects of Identification with the Student Community, Social Activity Spaces and Social Liability Interventions

**DOI:** 10.3390/ijerph20054511

**Published:** 2023-03-03

**Authors:** Brennan Davis, Cornelia Pechmann

**Affiliations:** 1Orfalea College of Business, California Polytechnic State University, San Luis Obispo, CA 93407, USA; 2Paul Merage School of Business, University of California Irvine, Irvine, CA 92697, USA

**Keywords:** fast food proximity to school, identification with school, social marketing message

## Abstract

US schools have fast-food restaurants nearby, encouraging student patronage, unhealthy consumption, and weight gain. Geographers have developed an activity space framework which suggests this nearby location effect will be moderated by whether people perceive the location as their activity space. Therefore, we study whether students perceive a fast-food restaurant near school as their activity space, and whether social marketing messages can change that perception. We conducted six studies: a secondary data analysis with 5986 students, a field experiment with 188 students, and four lab experiments with 188, 251, 178, and 379 students. We find that students who strongly identify with their student community patronize a fast-food restaurant near school (vs. farther away) because they view it as their activity space, while students who weakly identify do not. For example, in our field experiment, 44% vs. 7% of students who strongly identified with the student community patronized the near versus farther restaurant, while only 28% versus 19% of students who weakly identified patronized the near and farther restaurants comparably. We also find that to deter the strong identifiers, messages should convey that patronage is a social liability, e.g., portray student activism against fast food. We show that standard health messages do not change perceptions of restaurants as social activity spaces. Thus, to combat the problem of fast-food restaurants near schools causing unhealthy consumption, policy and educational interventions should focus on students who strongly identify with their student community and find ways to weaken their perceptions that fast-food restaurants near schools are their activity spaces.

## 1. Introduction

Research finds that retail nearness relates to retail patronage and product consumption [1] which we will refer to as the nearby location effect. Much of this work has studied the effects of near retail locations that sell unhealthy or risky products, such as fast food [2,3,4] or alcohol or tobacco [5,6,7]. In the US, 1 in 3 students are overweight and 1 in 5 are obese [8]. Strikingly, the majority of schools have a fast-food restaurant within a 1-mile radius [9,10], and 40% of students eat fast food daily [11,12]. Nearby location effects have frequently been found; students who have fast food near school (vs. not) have poorer diets [1,2,13,14,15] and are more likely to be overweight or obese [2,3,9,14,15,16]. Virtually all studies have been observational, not experiments, but the robust results are compelling. The problem with fast-food restaurants contributing to obesity is now a global one. US fast-food ad spending is increasing in non-US markets [17], and in China, fast-food sales and obesity rates are concurrently increasing [18].

In the US, high school students’ fast-food consumption is rising because there are more open campuses, meaning students can leave school for lunch, not just eat fast food before or after school [12,19,20]. Approximately 50% of California high schools have open campuses [21], and 67% in Oregon [12]. Moreover, research indicates that fast-food restaurants near schools have a disproportionate impact on minority and low-income students [3,14,15], because fast-food restaurants are more often situated by their schools [22,23], a situation that has worsened over time [23]. Low-income adults are also disproportionately affected by fast-food proximity [24]. 

The most common policy solution in the US has been to try to ban fast-food restaurants near schools [25,26,27] and otherwise restrict land use for fast-food restaurants [1]. Small affluent communities have had some success with locational bans, but urban, racially diverse communities where fast-food restaurants already abound have faced fierce business opposition dooming their efforts to limit fast food [25,26]. Very little research has tried to identify other solutions to the problem [28]. Moreover, research on fast-food proximity has lacked a unifying framework that, for instance, identifies relevant moderators and mediators.

In this research, we borrow a unifying framework from the geography literature which posts that the most fundamental predictor of any nearby location effect relates to whether people perceive it as their activity space [29,30]. According to the activity space literature, an unsupervised location near adolescents will emerge as their risky social activity space if their own peer group congregates there [31,32,33,34]. So, we asked the following question: When might a fast-food restaurant near school emerge as a social activity space for students, encouraging patronage? 

We reasoned that a nearby fast-food restaurant could become a social activity space for students who strongly identify with their student community. Due to their activities in and around school and their identification with their schoolmates, it could become a popular destination for these students to meet up. When students are strongly identified, this means they feel that they share beliefs, interests, and values with other students at their school, and feel accepted and liked by them [35,36,37,38,39]. Strong identification is often ignited when students engage in school-sponsored extracurricular activities such as sports, clubs, or the arts [40,41,42]. US schools provide wide access to extracurricular activities, and thus building strong identification with the student community is not limited to white or wealthy students [38,39,43].

Many US schools use surveys to measure students’ level of identification with the student community because of its predictive value [38,39,43]. Strong identification has been found to relate to many positive behaviors and to protect against numerous negative behaviors, from the teenage years through college [36,42,44]. Students who strongly identify with their student community tend to be more committed to academic goals [39,40,42] and less likely to use cigarettes, marijuana, or cocaine [41,42], though more likely to drink alcohol [40,41,45,46]. 

We posit that when a fast-food restaurant is located near school (vs. farther away), it will be perceived by high identifiers as their social activity space, attracting them there and promoting unhealthy eating. We will measure students’ perception of the location as their social activity space by asking them whether they go there to see friends. Strong identifiers should agree; weak identifiers should not. If the nearby fast-food restaurant is a draw for the strongly identified students, it is unlikely to attract the weakly identified, because different peer groups tend to hang out in separate places [29,30,31,32,34]. To summarize, we test the following hypothesis.

**H1.** 
*Among students who strongly identify with their student community, the location of a fast-food restaurant near (vs. farther from) school will enhance patronage because of a stronger perception that it is their social activity space. Among the weakly identified students, a fast-food restaurant near (vs. farther from) school will not have these effects.*


If students who are strongly identified with their student community think the fast-food restaurant near school is their social activity space, how might they be dissuaded? Studies have examined social marketing messages to deter students from unhealthy eating [47,48], alcohol use [49,50], and drug use [51,52,53]. Messages countering activity spaces have not been studied. However, the activity space framework posits that those spaces attract people by providing social benefits such as seeing friends [33,34,54]. Because the attraction is a social benefit, reducing its attraction will likely require reversing that perception to one of a social liability. Stating that the fast-food restaurant’s food is unhealthy is unlikely to be effective because it does not address students’ perception that it is a social activity space. An analogous situation occurs with smoking; adolescents start smoking for social acceptance [55,56]. To reduce its attraction, the opposite message must be conveyed: smoking is a social liability [53,57]. It is generally ineffective to convey to adolescents that smoking is a health liability [58]. 

It will be challenging to reverse adolescents’ perception that a previously acceptable hangout has become socially unacceptable among their peers. How can they be persuaded to see it differently? An emerging approach is to educate adolescents that marketers target them for unhealthy products and encourage student activism against being so targeted [59,60,61]. Sometimes students even engage in major activism, by which we mean they actively protest or boycott a product. A student-run product boycott is highly likely to make the product socially unacceptable to use among their peers. The US “truth” campaign against big tobacco did this effectively [61,62]. We tested activism messages and hypothesized the following.

**H2.** 
*Among students who strongly identify with their student community, the location of a fast-food restaurant near (vs. farther from) school, which normally attracts patronage, will no longer do so if a message conveys going there as a social liability. A health liability message will not have this effect. Among the weakly identified students, no such effects will be observed.*


## 2. Study 1 Materials, Methods, and Results

### 2.1. Overview

In Study 1, we used Geographic Information System (GIS) data on fast-food restaurant locations combined with California’s Healthy Kids student survey to study whether a fast-food restaurant near school (vs. farther away) increased students’ fast-food restaurant patronage. We sought to determine whether a nearby location effect mainly occurred among students who strongly identified with their student community. 

### 2.2. Participants

The participants were 5986 eleventh grade students who completed the long form of the California Healthy Kids Survey. They were the oldest respondents, most likely to have off-campus lunchtime privileges and make their own food decisions. Most participants were aged 16 (50%) or 17 (43%) and half were female (53%). They were ethnically diverse; 18% were Non-Hispanic White, 61% Hispanic, 29% Asian, 11% Black, 3% Hawaiian, and 8% Native American (2+ ethnicities could be chosen). Additionally, 59% were socioeconomically disadvantaged, eligible for free or reduced-price school meals due to low family incomes. 

### 2.3. Measures

To determine if at least one fast-food restaurant was near schools, we merged two types of GIS data: (1) the locations of all California high schools from the state’s Department of Education, and (2) the locations of all California fast-food restaurants from the GIS firm ESRI’s Business Analyst product using NAICS code 722513 [63,64]. The restaurant-to-school distance was the traversable distance, considering roads [65]. Research shows fast-food restaurants tend to be clustered in a one-mile radius around schools in the US [9,10], so we coded whether there was 1+ restaurant within one mile of each school.

To measure students’ fast-food restaurant patronage and identification with the student community, we used the California Healthy Kids school survey administered to public school students by the state’s Department of Education. Schools were sampled to represent the school district populations. Students were required to complete the survey if selected unless a parent actively withheld consent. We used surveys from 2011–2012 and 2013–2014, obtained GIS data for the same years, and verified result consistency across years. We used the long-form survey that was administered in 27 randomly selected public high schools (N = 222 students per school in average), because it asked: “How many times did you eat fast food in the past 24 hours?” (0 = 0 times; 5 = 5 or more times). It also included a measure of identification with the student community: “I feel like I am part of this school.”, “I feel close to the people at this school.”, and “I am happy to be at this school.” (1 = strongly disagree, 5 = strongly agree, averaged, α = 0.83). 

### 2.4. Analyses

We estimated a hierarchical ordinary least square regression model of fast-food restaurant patronage, relating it to a fast-food restaurant near school, identification with the student community, and their interaction [2]. We used a hierarchical model to account for student observations being non-independent [3]. This allowed us to test our main hypothesis (H1) at the individual level while controlling for some students being from the same county or the same school within the county. To assess the interaction between nearness and identification, we used floodlight analysis [66]. Though we used ordinary least squares because the dependent variable was a scale (e.g., 5 = 5 times or more), all models were re-estimated using Poisson regressions for count variables with similar results.

### 2.5. Results

In our sample of predominantly urban, ethnically diverse, and economically disadvantaged high school students, 94% (N = 5627) had a fast-food restaurant near school; 6% did not (N = 359). They reported consuming fast food 0.83 times (SD = 1.19) in the past 24 h, and their mean identification with the student community was 3.40 (SD = 0.94, 1–5 scale). Whether these students had a fast-food restaurant near their school, as opposed to all fast-food restaurants being relatively far from school, did not relate to their fast-food restaurant patronage as a main effect (b = 0.10, df = 5980, z = 0.85, *p* = 0.40). Students’ identification with their student community related negatively to their fast-food restaurant patronage as a main effect, indicating that strong identification generally had a protective effect (b = −0.16, df = 5980, z = 3.79, *p* < 0.001). Finally, as hypothesized (H1), there was an interaction between fast-food restaurant nearness to school and identification with the student community on restaurant patronage (b = 0.23, df = 5980, z = 2.81, *p* < 0.01). See Table 1. 

In supplemental analyses, we looked at whether students’ ethnicity or income (free or reduced-price meal eligible) related to identification with their student community. Hispanic (r = 0.03, *p* = 0.06) and White (r = 0.08, *p* < 0.001) correlated positively with identification. Black (r = −0.06, *p* < 0.001), Asian (r = −0.03, *p* < 0.01), and low income (r = −0.03, *p* < 0.05) correlated negatively. Native American (r = 0.005, *p* = 0.72), Pacific Islander (r = −0.01, *p* = 0.61), and mixed (r = −0.007, *p* = 0.58) were uncorrelated. However, all correlations were weak. We included ethnicity and income as covariates in our model, but the results were unaffected (see Appendix A).

We also conducted a floodlight analysis to understand the interaction effect we had observed [66]. Among students who strongly identified with their student community, fast-food restaurant patronage was higher if a restaurant was near school compared to farther away (right side of graph, solid line > dotted line). This nearby location effect was significant at an identification level of 4.25 or more on a 1–5 scale, *p* < 0.05 (shaded area on graph). Among students who weakly identified with their student community, fast-food restaurant patronage was higher if a restaurant was farther from school compared to nearby (left side of graph, dotted line > solid line). However, this effect only became significant at a very low identification level of 1.25 or less on a 1–5 scale, nearly at the scale endpoint. See Figure 1, which illustrates why the negative main effect for identification was qualified by the two-way interaction. The stronger the identification, the less the fast-food patronage if it was far from school (steep negative slope for dotted line); this was much less so if the fast food was near school (slight negative slope for solid line). 

### 2.6. Discussion

In Study 1, we analyzed data from a large statewide survey of high school students. We found that, overall, strong identification with the student community reduced the risk of fast-food restaurant patronage, consistent with other protective effects of strong identification. However, while the strongly identified students tended to avoid unhealthy fast food, when a fast-food restaurant was located near (vs. farther from) school, their patronage increased, indicating it was a social activity space for them. Weakly identified students showed elevated patronage overall, but no more so when a restaurant was near (vs. farther from) school and tending toward the reverse. Identification with the student community was not highly related to ethnicity or income, and our results were confirmed even when these variables were controlled. However, these data were observational, so unobserved variables could have affected the results. Additionally, while students reported how many times they ate fast food, we could not verify where the food came from or whom they were with. Additionally, the data were skewed toward fast-food being near schools due to our sample.

## 3. Study 2 Materials, Methods, and Results

### 3.1. Overview

For Study 2, we conducted a behavioral field experiment. Each student received a money-saving promotional coupon for the same fast-food restaurant (e.g., McDonalds) redeemable either at a location near school or, alternatively, farther away but still reachable. We then monitored actual coupon redemption at both locations. To manipulate identification, students completed an essay eliciting either strong or weak identification.

### 3.2. Design and Participants

The design was a 2 (restaurant nearness to school) × 2 (identification with the student community) between-subjects factorial with both factors manipulated. Participants were 153 older adolescents, university students, with a mean age of 20.5 years, 44% female, 75% White Non-Hispanic, 10% Hispanic, 9% Black, 12% Asian, and 3% other. Respondents could select more than one ethnicity. We recruited 188 but dropped 35 because they had dietary restrictions precluding fast food or did not complete the identification essay (N_near, strong_ = 30; N_near, weak_ = 43; N_far, strong_ = 37; N_far, weak_ = 43). 

### 3.3. Manipulations

Students participated for partial course credit. First, we manipulated their identification with the student community using an essay task [35]: “Please write for a few minutes (about 1 paragraph). In what ways do you think you are similar to (different from) other students here at University X? Consider attributes, interests, beliefs, values, etc. Try to recall some specific experiences that made you feel a part of (different from) the University X student community.” University X was named as their university in this and all studies. 

Next, fast-food restaurant nearness to school was manipulated by giving students one of two promotional coupons for the same fast-food restaurant, redeemable at one of its two locations, one near the school, the other farther away but still accessible because most students had cars, or their friends did. Each promotional coupon offered “$5 off any food item” and showed the location. The nearby (farther) location was described as “2 (20) minutes away” and was in fact about 0.5 (5) miles away, but we referred to drive time rather than miles because travel time more meaningfully conveys distances [67]. The coupon also showed the address and a small map and stated the $5 off promotion could only be used at that location on that day by 7 PM (see Appendix B). We asked participants not to share or discuss their coupon with others.

### 3.4. Measures

Research assistants were stationed at the two restaurant locations and collected the promotional coupons at the end of the redemption period. A subtle mark on each coupon identified each participant’s identification condition. We collected the promotional coupons from the cash registers, but we could not obtain the sales receipts. Therefore, we could not determine what participants bought or whom they were with if anyone. While having each individual sales receipt would have been more informative, the restaurants did not allow this, as it would have been obtrusive and slowed down their processes, which depend on speed.

The next day, participants completed an online survey with a restaurant nearness manipulation check which displayed their promotional coupon and asked: “How spatially close or far does this restaurant seem to you?” (very far to very close, very distant to very near, and very large travel time to very small travel time, 1–7, α = 0.91) [68]. A product attitude covariate was measured: “I like [restaurant X, named]” with 1 = strongly disagree and 100 = strongly agree, to control for product disinterest [69]. Demographics were measured in all studies.

To check the identification essay manipulation, two raters blinded to condition read each essay and answered [35]: “To what extent does this individual … seem to identify with University X?” “… discuss themselves as a part of the University X community?” “…discuss themselves as similar to other University X students?” “… discuss themselves as a prototypical University X student?” “… seem to feel that being a part of University X is important to them?” (1 = not at all, 6 = a great deal). Inter-rater reliability was high (α = 0.84).

### 3.5. Analyses

Restaurant patronage data were analyzed using 2 (nearness) × 2 (identification) logistic regressions as the outcome was binary (1 = redeemed coupon, 0 = did not redeem) and interactions were assessed using planned pairwise comparisons. Manipulation check data were analyzed similarly but using ANOVAs. In this and all lab studies, we included the product attitude covariate in our models and report covariate adjusted values.

### 3.6. Manipulation Check Results

Students who received the promotional coupon for the fast-food restaurant location that was near versus farther from school reported its location as nearer (F(1, 148) = 71.49, *p* < 0.001; M = 5.42 vs. 3.49), with no main effect for identification (*p* = 0.28), no interaction (*p* = 0.53), and no effect for the product attitude covariate (*p* = 0.35). The raters judged the essays designed to elicit strong as compared to weak identification as more strongly identifying with the student community (F(1, 148) = 108.64, *p* < 0.001; M = 5.40 vs. 2.66), with no main effect for restaurant nearness (*p* = 0.43), no interaction (*p* = 0.89), and no effect for the product attitude covariate (*p* = 0.58). 

### 3.7. Main Results

On fast-food restaurant patronage, while there was no main effect for identification (b = −0.12, z(148) = 0.53, *p* = 0.60), there was a main effect for restaurant nearness (b = 0.77, z(148) = 3.33, *p* = 0.001), but it was qualified by an interaction between restaurant nearness and identification (b = 0.50, z(148) = 2.18, *p* = 0.03), and the product attitude covariate also related to patronage (b = 0.04, z(148) = 2.85, *p* = 0.004). Students who strongly identified with the student community patronized the near versus farther restaurant more (44% vs. 7%; b = 1.27, z(148) = 3.46, *p* = 0.001), while students who weakly identified patronized the near and farther restaurants comparably (28% vs. 19%; b = 0.26, z(148) = 0.94, *p* = 0.35). See Figure 2.

### 3.8. Discussion

In Study 2, we gave students a promotional coupon for a fast-food restaurant that was redeemable at only one location, either near or farther from school, and we observed coupon redemption. We also manipulated their identification with the student community. We found direct behavioral evidence that, among those who felt strongly identified with the student community, a fast-food restaurant near versus farther from school was a significant draw. Students who felt weakly identified with the student community were equally likely to redeem the coupon irrespective of restaurant location. 

## 4. Study 3 Materials, Methods, and Results

### 4.1. Design and Participants

In Study 3, we investigated the underlying mediating process that may have caused students who were strongly identified with their student community to patronize a nearby fast-food restaurant. The posited mediator was the perception of it being a social activity space, i.e., where friends could be found. The design was a 2 (restaurant nearness to school) × 2 (identification with the student community) between-subjects factorial with nearness manipulated and identification measured. Participants were 188 older adolescents, university students, with a mean age of 19.5 years, 54.8% female, 63% White Non-Hispanic, 4% Hispanic, 16% Asian, 1% Black, and 16% other (2+ ethnicities could be selected). We recruited 198 but dropped 10 with dietary restrictions precluding fast food (N_near_ = 94; N_far_ = 94). 

### 4.2. Manipulations and Measures

Students completed a study “about consumer response to retailers” for partial course credit. Restaurant nearness was manipulated as follows: “Imagine you are just leaving the University X campus. You receive a text from a new donut shop at least 2 (20) minutes away from campus offering you an attractive promotional discount for donuts today only. Picture this place and the people there in your mind.” Then, we asked their restaurant patronage intent: “Would you redeem this promotional coupon to eat at the restaurant?” (1 = definitely not, 7 = definitely yes). Next, we performed a nearness manipulation check: “How spatially close or far does this restaurant seem to you?” (very far to very close, very distant to very near, and very large travel time to very small travel time, 1–7, α = 0.99). Then, we measured the mediator, the perception the fast-food restaurant was a social activity space to see friends: “Please indicate which items were salient to you when you decided whether to go eat at the restaurant”: “See friends” “Going to a place for people like me” “Be with people with whom I identify” (1 = strongly disagree to 7 = strongly agree; α = 0.73). 

After this, we measured identification with the student community (see Appendix C). We showed increasingly overlapping circles labeled “You” and “University X” (Tropp and Wright 2001) and asked: “Please click on the picture below that best describes how much you happily feel a part of your University X student community” (0 = no overlap, 7 = complete overlap).” Finally, we measured the product attitude covariate: “I like donuts” (1 = strongly disagree to 5 = strongly agree). The data were analyzed using 2 (nearness) × 2 (identification) ANOVAs, with interactions assessed using spotlight analysis [66]. Moderated mediation models used Hayes model 8 [70] with 5000 replications.

### 4.3. Manipulation Check Results

Students in the near versus farther restaurant condition reported the restaurant was nearer (F(1, 183) = 273.26, *p* < 0.001; M = 5.58 vs. 3.05). There was no main effect for identification (*p* = 0.56), no interaction (*p* = 0.27), and no effect of the product attitude covariate (*p* = 0.53). 

### 4.4. Main Results

We observed the hypothesized interaction between restaurant nearness and identification on restaurant patronage intent (F(1, 183) = 4.87, *p* = 0.03) which qualified the main effect for restaurant nearness that favored the near versus farther restaurant (F(1, 183) = 18.94, *p* < 0.001; M = 4.17 vs. 2.95), with no main effect for identification (F(1, 183) = 0.18, *p* = 0.67). Students strongly identified with their student community (mean + 1 SD) were more likely to intend to patronize the near versus farther restaurant (M = 4.54 vs. 2.58; t(183) = 5.32, *p* < 0.001). Students weakly identified with their student community (mean—1 SD), were indifferent to the near versus farther restaurant (M = 3.80 vs. 3.32; t(183) = 0.97, *p* = 0.33). The product attitude covariate also related to patronage (F(1, 183) = 25.08, *p* < 0.001). 

### 4.5. Results on Mediation

We observed a marginal interaction between restaurant nearness and identification on the posited mediator: students’ perception of the restaurant as their social activity space (F(1, 183) = 2.87, *p* = 0.09), with no main effect for nearness (F(1, 183) = 0.78, *p* = 0.38) but a main effect for identification (F(1, 183) = 5.17, *p* = 0.02). Students who strongly identified with their student community (mean + 1 SD) were more likely to perceive the near versus farther restaurant as their social activity space (M = 4.21 vs. 3.66; t(183) = 2.21, *p* = 0.03); while students who weakly identified (mean—1 SD) perceived the near and farther restaurants comparably (M = 3.83 vs. 4.04; t(183) = 0.64, *p* = 0.52). The product attitude covariate was unrelated to this perception (F(1, 183) = 1.53, *p* = 0.22).

In a direct test of mediation, among students who strongly identified with their student community (mean + 1 SD), the effect of the near versus farther restaurant on patronage intent was mediated by the perception the restaurant was their social activity space (indirect effect = 0.18, SE = 0.11, 95% CI = 0.01, 0.46). Among students who weakly identified with their student community (mean—1 SD), there was no such effect (indirect effect = −0.01, SE = 0.09, 95% CI = −0.22, 0.16). See Figure 3.

### 4.6. Discussion

In Study 3, we manipulated fast-food restaurant nearness to school, and we measured students’ identification with their student community and the theorized mediator. We found direct evidence of mediation. Students who strongly identified with their student community said they decided to patronize the nearby (vs. farther away) fast-food restaurant to see friends, indicating they perceived it to be their social activity space. Students who weakly identified with their student community did not perceive it this way or patronize it. 

## 5. Study 4 Materials, Methods, and Results

### 5.1. Design and Participants

Study 4 tested a mild form of student activism: a disparaging social media post from a student at the high school, indicating it would be a social liability to be seen at a fast-food restaurant. The design was a 2 (restaurant nearness to school) × 2 (identification with the student community) × 2 (social liability vs. control message) between-subjects factorial, with all three factors manipulated. Participants were 251 older high school students from MTurk, screened to be in high school but over the age of 17 to exclude minors as mandated by our human subjects review board. Virtually all were aged 18, 35% were female, 67% White Non-Hispanic, 15% Asian, 11% Hispanic, 8% Black, and 3% other (permitting 2+ ethnicities). We recruited 273 but dropped 22 who did not complete the identification manipulation (N_near_strong_control message_ = 31; N_near_strong_social liability message_ = 41; N_far_strong_control message_ = 28; N_far_strong_social liability message_ = 31; N_near_weak_control message_ = 34; N_near_weak_social liability message_ = 20; N_far_weak_control message_ = 30; N_near_weak_social liability message_ = 36).

### 5.2. Manipulations and Measures

The social liability message, described as a social media post from a student at their high school, stated: “Students at this school would never be seen by friends at fast-food restaurants.” The control message was likewise described as a social media post from a student at their high school: “The school library is now open on weekends.” These social media posts were displayed on mobile phones (see Appendix D). Then, we used our prior methods to manipulate identification via an essay task, manipulate fast-food restaurant nearness using a burger restaurant (2 vs. 20 drive-time minutes) and measure restaurant patronage intent (“Would you redeem this promotional coupon…”). 

We used our prior nearness manipulation check (α = 0.95). We used an identification manipulation check with increasingly overlapping circles “You” and “High School X” [37] that asked: “Please click on the picture below that best describes how much you feel part of [or close to, or happily part of] your High School X student community” (0 = no overlap, 7 = complete overlap, α = 0.94). Our manipulation check of the social liability message measured seeing that post (1 = strongly disagree, 5 = strongly agree), e.g., “Students at my high school would not like to be seen by friends at fast-food restaurants” (3 items, α = 0.82). Finally, we measured the product attitude covariate (“I like fast food” 1 = strongly disagree, 5 = strongly agree; 4 missing responses). The data were analyzed using 2 (nearness) × 2 (identification) × 2 (message) ANOVAs and interactions were assessed using planned pairwise comparisons. 

### 5.3. Manipulations Check Results

Students in the near versus farther condition reported the fast-food restaurant was nearer to them (F(1, 238) = 32.35, *p* < 0.001; M = 5.16 vs. 4.02). Those in the strong versus weak identification condition reported more identification (F(1, 238) = 46.93, *p* < 0.001; M = 4.00 vs. 2.74). Those seeing the social liability versus control message reported what it said; students would not like to be seen at fast-food restaurants (F(1, 238) = 12.92, *p* < 0.001; M = 3.12 vs. 2.97). There were no other effects.

### 5.4. Restaurant Patronage Intent

As hypothesized (H2), there was a three-way interaction on restaurant patronage intent (F(1, 238) = 6.89, *p* = 0.009). There were no other effects except a main effect for the product attitude covariate (F(1, 238) = 22.70, *p* < 0.001). With the control message, strongly identified students increased their intent to patronize a fast-food restaurant if near versus farther from school (t(238) = 10.51, *p* = 0.009; M = 5.48 vs. 4.10); weakly identified students did not (t(238) = 0.07, *p* = 0.79; M = 4.85 vs. 4.95). With the social liability message, strongly identified students no longer increased their intent to patronize if near versus farther (t(238) = 0.91, *p* = 0.34; M = 4.87 vs. 5.24), and weakly identified students remained indifferent to nearness (t(238) = 0.57, *p* = 0.45; M = 4.94 vs. 4.60). See Figure 4.

### 5.5. Discussion

We tested a mild form of student activism; a student posted that going to a nearby fast-food restaurant was a social liability. Others saw a control post. The effects were, again, limited to the strongly identified students. If they saw the control post, they were attracted to a nearby (vs. farther) fast-food restaurant; if they saw the social liability post, they were not. 

## 6. Study 5 Materials, Methods, and Results

### 6.1. Design and Participants

We tested a stronger student activism message; students announced a boycott of nearby fast-food restaurants. Student activism of this type is increasingly prevalent; thus, the message was realistic [71]. The design was a 2 (restaurant nearness to school) × 2 (social liability vs. control message) between-subjects factorial with both factors manipulated. All participants were manipulated to feel strongly identified with their student community. We studied 178 older adolescents, university students, with a mean age of 19.7 years, 68% female, 60% White Non-Hispanic, 6% Hispanic, 1% Black, 20% Asian, and 14% other (2+ ethnicities allowable). We recruited 193 but dropped 15 because of dietary restrictions precluding fast food or the identification manipulation not being done (N_near, control message_ = 51; N_near, social liability message_ = 41; N_far, control message_ = 40; N_far, social liability message_ = 46). 

### 6.2. Manipulations and Measures

The social liability message was a color poster of students stating: “The University X student community boycotts fast food near campus.” The visually identical control message stated: “The University X student community boycotts tobacco shops near campus.” (See Appendix E). We used our prior strong identification essay and manipulated nearness as 2 vs. 20 drive-time minutes. We used prior measures of restaurant patronage intent, the nearness manipulation check (α = 0.98), identification with the student community (α = 0.86), and the product attitude covariate. The social liability manipulation check asked whether “the poster encouraged the University X student community to boycott fast-food restaurants” (1 = strongly disagree to 5 = strongly agree). Data were analyzed using 2 (nearness) × 2 (message) ANOVAs and interactions using planned pairwise comparisons. 

### 6.3. Manipulations Check Results

Students in the near versus farther condition reported the restaurant was nearer (F(1, 173) = 65.34, *p* < 0.001; M = 5.62 vs. 3.39). Students who saw the social liability versus control message reported the content correctly (F(1, 173) = 341.84, *p* < 0.001; M = 4.30 vs. 1.33). Identification was strong as manipulated (M = 4.33 out of 5). There were no other effects.

### 6.4. Main Results

Restaurant nearness and message interactively affected restaurant patronage intent (F(1, 173) = 3.89, *p* < 0.05) which qualified main effects for near versus far (F(1, 173) = 17.20, *p* < 0.001, M = 4.16 vs. 3.11) and social liability versus control message (F(1, 173) = 11.81, *p* < 0.001, M = 3.18 vs. 4.10), with product attitude covariate having no effect (F(1, 173) = 2.08, *p* = 0.15). When the strongly identified students saw the control message, as before, they reported higher intent to patronize the near versus farther fast-food restaurant (t(173) = 19.03, *p* < 0.001; M = 4.75 vs. 3.32), but when they saw the social liability message, this effect was nullified (t(173) = 2.32, *p =* 0.130; M = 3.44 vs. 2.91; see Figure 5).

### 6.5. Discussion

In Study 5, we showed strongly identified students a forceful activism message: a boycott against nearby fast-food restaurants, implying that going there would be a social liability. The strong identifiers who saw the control message were attracted to the nearby (versus farther) fast-food restaurant; those who saw the social liability message no longer were. 

## 7. Study 6 Materials, Methods, and Results

### 7.1. Design and Participants

Study 6 tested a health liability message stressing that fast food was unhealthy. The design was a 2 (restaurant nearness to school) × 2 (identification with the student community) × 2 (health liability versus control message) between-subjects factorial, with all three factors manipulated. Participants were 379 older adolescents, university students, with a mean age of 20.3 years, 48% female, 64% White Non-Hispanic, 12% Hispanic, 21% Asian, 1% Black, and 3% other. We recruited 425 but dropped 46 who had dietary restrictions precluding fast food; all completed the identification manipulation (N_near, strong, control message_ = 46; N_near, strong, health liability message_ = 45; N_far, strong, control message_ = 40; N_far, strong, health liability message_ = 57; N_near, weak, control message_ = 59; N_near, weak, health liability message_ = 45; N_far, weak, control message_ = 46; N_far, weak, health liability message_ = 41).

### 7.2. Manipulations and Measures

The health liability message was visually similar to the liability message used in Study 5. This message showed a similar group of students who proclaimed: “Students at this school do not like unhealthy fast-food restaurants.” Thus, the main emphasis was unhealthy food. The control message was: “The school library is now open on weekends.” (See Appendix F). We used our prior manipulations of identification and nearness.

Next, we asked: “Would you make a purchase at this fast-food restaurant?” (1 = extremely unlikely, 7 = extremely likely). We checked our nearness manipulation as before (α = 0.98), and our identification manipulation (“I identify with the University X student community.” 1 = strongly disagree, 7 = strongly agree). We checked our message manipulation: “This study showed me a poster discouraging University X students from going to fast-food restaurants” (1 = strongly disagree to 5 = strongly agree). We measured the product attitude covariate as before. Data were analyzed using 2 (nearness) × 2 (identification) × 2 (health liability versus control message) ANOVAs and interactions were assessed using planned pairwise comparisons. 

### 7.3. Manipulations Check Results

Students in the near versus farther condition reported the restaurant was nearer (F(1, 370) = 323.83, *p* < 0.001; M = 5.34 vs. 3.21). Those in the strong versus weak identification condition reported stronger identification (F(1, 370) = 6.54, *p* = 0.01; M = 5.10 vs. 4.83). Those shown the health liability versus control message reported the content correctly (F(1, 370) = 353.23, *p* < 0.001; M = 3.59 vs. 1.50). There were no other effects.

### 7.4. Main Results

There was a three-way interaction for nearness, identification, and health liability message on restaurant patronage intent (F(1, 370) = 4.97, *p* = 0.03), a main effect for nearness (F(1, 370) = 49.48, *p* < 0.001), a main effect for the product attitude covariate (F(1, 370) = 40.55, *p* < 0.001), but no other effects. The control message results replicated what we saw earlier. Those strongly identified with the student community reported higher intent to patronize the fast-food restaurant when near versus farther from school (t(370) = 4.03, *p* < 0.001; M = 4.48 vs. 3.10); weakly identified students did not (t(370) = 1.67, *p* = 0.10; M = 4.04 vs. 3.53). The health liability message results were different. This message failed to lower the attraction of nearby fast food among strong identifiers, and it increased fast-food attraction among weak identifiers. After seeing the health liability message, the strong identifiers continued to report higher intent to patronize the near (vs. farther) fast-food restaurant (t(370) = 3.35, *p* < 0.001; M = 4.31 vs. 3.26). The weak identifiers did likewise, primarily because they became attracted to the near restaurant (t(370) = 4.85, *p* < 0.001; M = 4.62 vs. 2.96; see Figure 6).

### 7.5. Discussion

Study 6 tested a health liability message, stressing that fast food was unhealthy. Showing it to strong identifiers had no effect; a fast-food restaurant near (vs. farther from) school continued to increase patronage intent. Showing it to weak identifiers was counterproductive; now even they were attracted to the nearby (vs. farther) restaurant. Conceivably, the weak identifiers experienced reactance when told the food was unhealthy, and so they decided to patronize the nearby restaurant.

## 8. Final Discussion

### 8.1. Contributions

Fast-food restaurants near schools are problematic, contributing to poor diet, weight gain and obesity. Our novel hypothesis, supported by detailed data analysis, is that teenagers who have a strong sense of identity with their student community, although their risk is usually low, face the greatest risk of a fast-food restaurant near school, because they think the restaurant is their social activity place. Advocating policy and educational interventions to change this view has important practical significance for solving the problem of unhealthy consumption caused by fast-food restaurants near schools. 

Our findings suggest new policy approaches to addressing the problem of unhealthy fast-food restaurants near schools, that are not reliant on zoning restrictions that have been tried in the past [1]. Zoning restrictions have largely been unsuccessful in the US, especially at protecting disadvantaged students [25,26]. We advocate the use of school policies, social marketing messages, and educational efforts targeted at students that seek to change their perception of fast-food restaurants near school from socially beneficial spaces to social liability spaces. 

Specifically, we recommend that teachers use their educational toolbox to encourage student activism, e.g., boycotts against local fast-food restaurants. Local activism is an increasingly popular strategy for promoting social change in the US and abroad, used by students [71], employees [72,73], even corporations [74]. For instance, social studies, nutrition, or language teachers might encourage students to think critically about whether and how they have been targeted by unhealthy fast-food restaurants. If students understand they have been targeted by fast-food marketers who have encouraged them to eat unhealthy food since they were small children unable to think critically, they may want to do something, perhaps start a boycott. 

### 8.2. Links to Past Literature

Our research complements past work that discovered that student demographics moderate their vulnerability to fast-food restaurants near schools [3,14,15]. We study a different moderating variable, not a demographic variable, but rather strong identification with the student community [38,39,43]. Strong identification generally protects students from risk [36,42,44], but in the case of fast food it elevates their risk because they perceive a fast-food restaurant near school as a social activity space where they can derive social benefits, i.e., see friends. Policymakers should adopt educational and messaging strategies to change this perception, so that going to a fast-food restaurant is a social liability. They should not stress the health liability, i.e., unhealthy food, as we found this to be ineffective.

Our work supports the geographers’ activity space framework which indicates that the most fundamental moderator of any nearby location effect relates to whether people perceive that location as their activity space [29,30,31,32]. However, we add to the work in geography by showing that students’ identification with their school community affects their activity space perceptions. 

We also contribute to past work in marketing on social influences that often adversely affect food consumption. Studies have found that people tend to match others’ portion sizes despite their hunger [75], match others’ menu selections despite their preferences [76] and use food to signal preferred identities independent of other considerations [77]. We demonstrate another adverse social influence on food consumption by showing that adolescents will go to a fast-food restaurant, despite its unhealthy food, to see friends.

### 8.3. Theoretical and Methodological Contributions

Our work makes a theoretical contribution by showing that an individual difference variable, student identification with their student community, moderates their perception of whether they will see peers at a nearby fast-food restaurant and want to go there. Past research tells us that adolescents’ perceptions of peers strongly influence their use of drugs and alcohol [55,78,79], elevating social concerns over health ones [58,80,81]. We add the insight that peers also matter with fast food. While this may seem to be a logical extension, the focus of past fast-food research has been on restaurant location not peer perceptions. Activity space geographers have challenged the narrow focus on location, noting that perceptions of locations as activity spaces also matter [31,32,33,34]. However, we are the first to identify an individual difference variable, adolescent identification with the student community, which affects activity space perceptions. Moreover, we demonstrate how to measure activity space perceptions as a mediating variable, and how to test for mediation.

### 8.4. Limitations

Limitations of our research include that we focused on fast-food restaurant patronage not consumption. We do not know what students might have eaten at the restaurants and, thus, it is conceivable some might have chosen the relatively healthier items. We did not study socializing at the restaurants, only whether students decided to go to see friends. Only one of our studies (study 3) measured the theorized mediating process, about the nearby restaurant being a social activity space or hangout for friends. We used high school students in Studies 1 and 4, but otherwise used college students. Our findings replicate with both groups, consistent with extensive research indicating that adolescence extends from the teenage years through to about age 24 [79]. The entire period of adolescence is characterized by highly salient social goals and affiliation needs, and a tension between necessary dependence on parents versus independence from them, e.g., with respect to cars, meals, and privileges [82]. However, as the younger adolescents are understudied, more research should be done on them.

We also recommend studies of other risky locations near schools, to ascertain if students who are strongly identified with their student community are especially vulnerable. What about nearby liquor, tobacco, nicotine vape or pot (cannabis) retailers; do they attract students who are strong identifiers? What about nearby fitness centers or fresh produce markets (farmers markets); do they attract strong identifiers but elicit positive behaviors? In addition, researchers should examine adults with workplaces nearby fast-food restaurants who vary in workplace identification, to see if the results replicate. Activity space researchers have replicated their findings among adolescents and adults, and replication work would be beneficial here too. Researchers should study other negative behaviors that might be evoked by strong identification with a student community, e.g., aggressive behavior at intercollegiate sports events. Among geography researchers, it would be useful to study other individual difference variables that may affect perceptions of locations as social activity spaces. “Location, location, location” is indeed important, but social perceptions of locations matter too and should be investigated further.

## 9. Conclusions

Fast-food restaurants near schools are problematic, contributing to poor diet, weight gain, and obesity. Adolescents who strongly identify with their student community, while generally at lower risk, face the greatest risk from fast-food restaurants near school because they perceive the restaurants as their social activity spaces. Education and policy should be directed at changing that perception. Students must perceive the restaurants differently, as well as adults.

## Figures and Tables

**Figure 1 ijerph-20-04511-f001:**
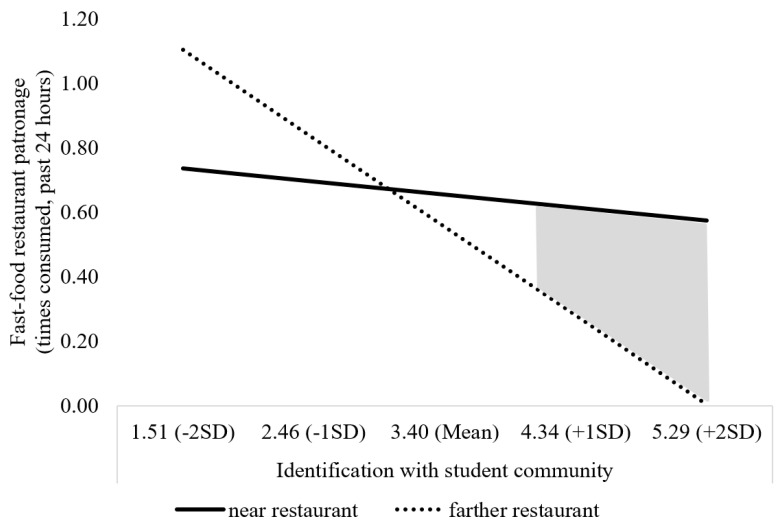
Study 1 Fast-food Restaurant Patronage due to Nearness to School and Identification. Note: Johnson–Neyman turning point = 4.25, *p* < 0.05.

**Figure 2 ijerph-20-04511-f002:**
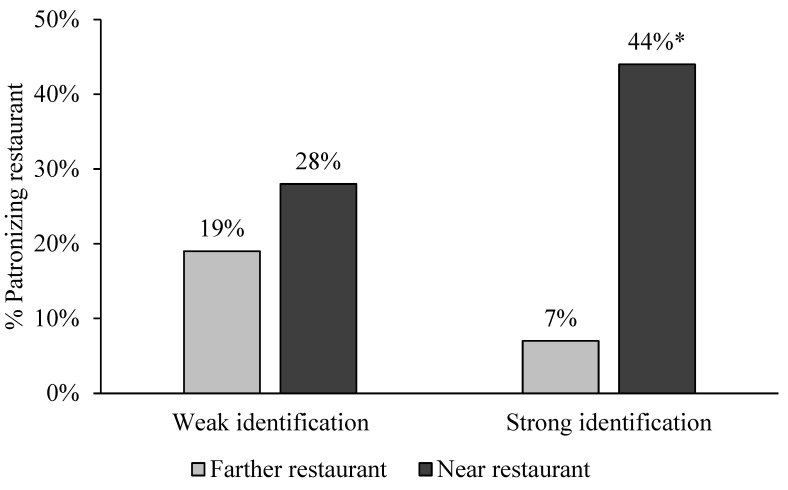
Study 2 Fast-food Restaurant Patronage due to Nearness and Identification. Note: * *p <* 0.001 comparing 44% with 7% for the strong identifiers.

**Figure 3 ijerph-20-04511-f003:**
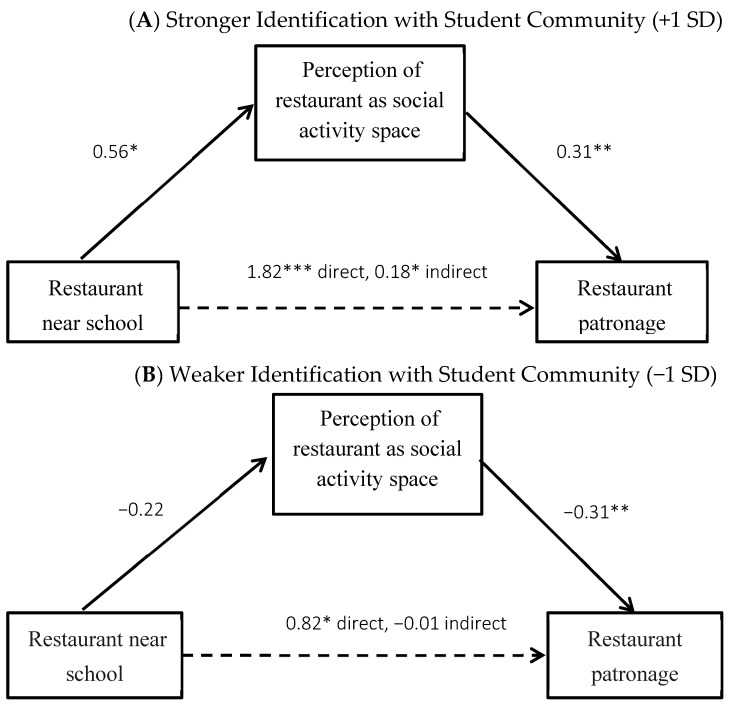
Study 3 Fast-food Restaurant Patronage Mediated by Activity Space Perceptions. Note: * *p* < 0.05, ** *p* < 0.01, *** *p* < 0.001.

**Figure 4 ijerph-20-04511-f004:**
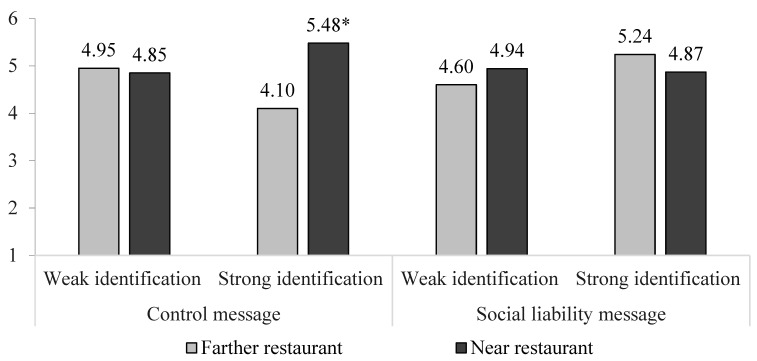
Study 4 Fast-food Restaurant Patronage after a Social Liability Message. Note: * *p* < 0.01 comparing 5.48 vs. 4.10 for the strong identifiers seeing a control message.

**Figure 5 ijerph-20-04511-f005:**
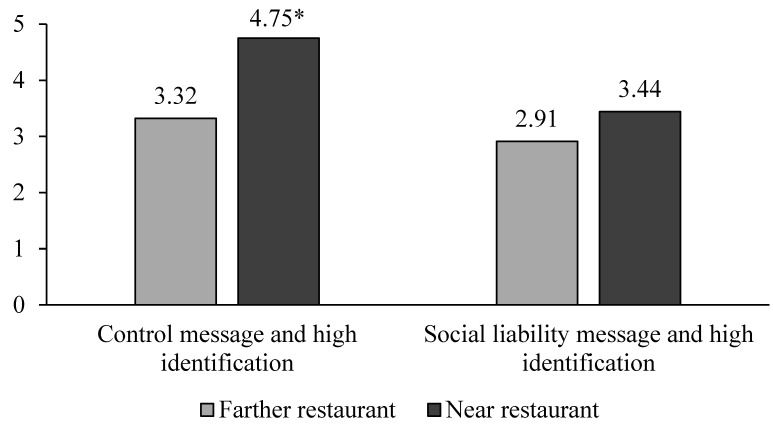
Study 5 Fast-food Restaurant Patronage after a Social Liability Message. Note: * *p <* 0.001 comparing 4.75 vs. 3.32 for the strong identifiers seeing a control message.

**Figure 6 ijerph-20-04511-f006:**
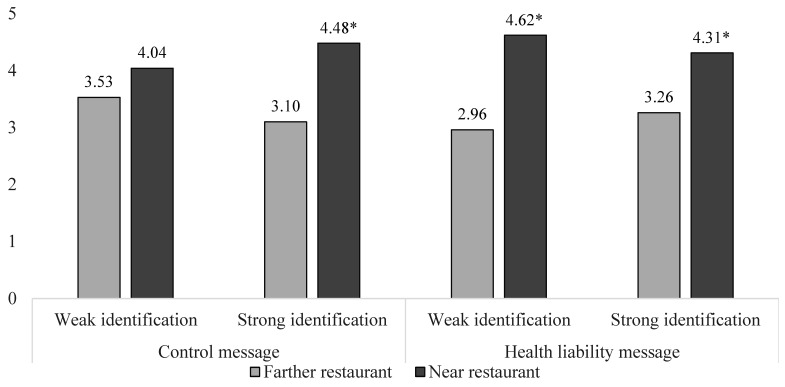
Study 6 Fast-food Restaurant Patronage Intent after a Health Liability Message. Note: * *p <* 0.001 for all comparisons shown.

**Table 1 ijerph-20-04511-t001:** Study 1 Fast-food Restaurant Patronage due to Nearness and Identification.

Predictor Variable	Relationship to Fast-food Restaurant Patronage *
Fast-food restaurant near school	b = 0.10, df = 5980, z = 0.85, *p* = 0.40
Identification with student community	b = −0.16, df = 5980, z = 3.79, *p* < 0.001
Near school × identification	b = 0.23, df = 5980, z = 2.81, *p* < 0.01
School (hierarchical model level 2)	b = 0.02, df = 5980, SE = 0.01, 95% CI 0.01, 0.06
County (hierarchical model level 3)	b = 0.25, df = 5980, SE = 0.11, 95% CI 0.10, 0.61

Note: * Patronage question asked how many times fast food was consumed in the past 24 h.

## Data Availability

The data presented in Studies 2–6 are available from either author upon request. Data from Study 1 were purchased from CHKS with a MOU that disallows data sharing because they must be purchased from CHKS.

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
