# Peer review of "When Students Patronize Fast-Food Restaurants near School: The Effects of Identification with the Student Community, Social Activity Spaces and Social Liability Interventions"

_ijerph, 2023, doi:10.3390/ijerph20054511_

Round 1

Reviewer 1 Report

This manuscript written by Brennan Davis  and Cornelia (Connie) Pechmann is a very excellent research on “When Students Patronize Fast-food Restaurants Near School:The Effects of Identification with the Student Community, Social Activity Spaces and Social Liability Interventions”.  I would like to say that it is is only acceptable after minör revision.

Author Response

Responses to Reviewer 1

 Thank you for your assessment that “this manuscript written by Brennan Davis and Cornelia (Connie) Pechmann is very excellent research on ‘When Students Patronize Fast-food Restaurants Near School: The Effects of Identification with the Student Community, Social Activity Spaces and Social Liability Interventions.’”

Thank you for the chance to submit a minor revision. We repeat your comments below and respond to them one at a time.

Reviewer comment: The questionnaire applied in this research has a very comprehensive number of participants. For this reason, the number of participants should be especially emphasized in the abstract section. This will make the study more impressive.

Author response: Thank you for this excellent suggestion. We have updated the abstract accordingly.

Reviewer comment: In addition, a brief summary of the findings (especially the percentiles) should be briefly highlighted in the abstract and conclusion section.

Author response: Thank you for this suggestion. We have updated the abstract to include the percentages from our field experiment.

Reviewer comment: In the introduction section, the related work has been written in great detail, but the cited literature is quite past tense, for example; 3 (2013); 13(2020); 14 (2015); 18 (2012); 19(2017); 20 (2009). It would be more beneficial to include the research findings of recent years.

Author response: We have updated the manuscript to include newer articles in the area including Marsh et al. 2021, Wheeler et al. 2022, Prescott et al. 2022, Hargreaves et al. 2022, Russo et al. 2022, and Wong et al. 2019. But we retained the classic articles in the area and cites to our own related articles. Thank you for this excellent suggestion.

Reviewer comment: One of the most frequently encountered problems in similar studies and survey applications is the evaluation of restaurants as fast food, and the criteria taken as reference should be questioned. In this study, you classified which of the food groups and evaluated them as fast food. Do you have any additional literature on this?

Author response: We added new explanations to the manuscript for how we classified restaurants as fast food: we used NAICS code 722513 in ESRI and have cited papers with the same: Russo et al. 2022 and Wong et al. 2019.

Reviewer comment: Likewise, it is very important that it is a study that covers restaurants that are 141 miles away from schools, and it would be useful to emphasize this in the abstract and conclusion section.

Author response: We were not sure what was meant with this suggestion. We do not mention restaurants 141 miles away from schools. We are happy to consider this point but respectfully request more clarification.

Reviewer comment: In the final discussion section, there are many studies similar to the findings and hypothesis of this research. The hypothesis of your research, which is different from other literature, should be explained in detail. Also summarize your innovative hypothesis of this research in the conclusion section. Extensive research but lacking innovative hypothesis.

Author response:  We have rewritten our final discussion to highlight our innovative hypothesis and separately to state the links between our research and past research. Thank you for this excellent suggestion.

 Reviewer comment: English language and style are fine/minor spell check required.

 Author response: Thank you. We have checked the manuscript for spelling and grammar errors and believe everything has been corrected.

Reviewer comment: Plagiarism ratio is quite successful.

Author response: We are happy to hear that the paper has a low plagiarism ratio.

Reviewer 2 Report

Fast-food restaurants by schools are problematic, contributing to poor diet, weight gain and obesity. Teenagers who have a strong sense of identity with the student group, although the risk is usually low, they face the greatest risk in the fast-food restaurant near the school, because they think the restaurant is their social activity place. Advocating policy and educational intervention to change this view has important practical significance for solving the problem of unhealthy consumption caused by fast-food restaurants near schools. This article has a novel idea and detailed data analysis, which can support the expressed views to a large extent. It is recommended to accept it after major revesion.

Author Response

Response to Reviewer 2

 Reviewer comment: Fast-food restaurants by schools are problematic, contributing to poor diet, weight gain and obesity. Teenagers who have a strong sense of identity with the student group, although the risk is usually low, they face the greatest risk in the fast-food restaurant near the school, because they think the restaurant is their social activity place. Advocating policy and educational intervention to change this view has important practical significance for solving the problem of unhealthy consumption caused by fast-food restaurants near schools. This article has a novel idea and detailed data analysis, which can support the expressed views to a large extent. It is recommended to accept it after major revision.

Author response: We have included your summary of our contributions in our final discussion to address a request made by Reviewer 1. We hope this is ok with you! We have made other major revisions as requested. We thank you for your positive and constructive comments.

Reviewer 3 Report

The paper conducted empirical analyses of the effect of students' community involvement and the location of fast-food restaurants on students' consumption at those restaurants. Although the paper appears to be of great interest to the majority of International Journal of Environmental Research and Public Health readers, I am concerned with the following issues. If necessary, the authors may need to make revisions.

According to my understanding, the authors measured students' fast food restaurant patronage by the number of times they ate at a fast-food restaurant within the previous 24 hours. The number of meals available at the fast-food restaurant can be 0, 1, 2, 3, 4, or 5 times. If the authors conducted a hierarchical regression with the number of meals as the dependent variable, does this imply they utilized a hierarchical Poisson regression or a hierarchical ordinal logistic regression? Please provide clarification.

Every study should have limitations, but the text does not specify the limitations. To indicate which topics should be studied in the future, it would be necessary to identify the limitations of this paper.

Author Response

Response to Reviewer 3

Reviewer comment: The paper conducted empirical analyses of the effect of students' community involvement and the location of fast-food restaurants on students' consumption at those restaurants. Although the paper appears to be of great interest to the majority of International Journal of Environmental Research and Public Health readers, I am concerned with the following issues. If necessary, the authors may need to make revisions.

 Author response: We are happy that our paper may be of great interest to the majority of International Journal of Environmental Research and Public Health readers. We have addressed your concerns below.

Reviewer comment: According to my understanding, the authors measured students' fast food restaurant patronage by the number of times they ate at a fast-food restaurant within the previous 24 hours. The number of meals available at the fast-food restaurant can be 0, 1, 2, 3, 4, or 5 times. If the authors conducted a hierarchical regression with the number of meals as the dependent variable, does this imply they utilized a hierarchical Poisson regression or a hierarchical ordinal logistic regression? Please provide clarification.

Author response:
The dependent variable in the first study is measured as “How many times did you eat fast food in the past 24 hours?” The response options were 0, 1, 2, 3, 4 or 5 or more times. Since this is technically a scale rather than count, and to be consistent with past work, we prefer to use hierarchical ordinary least squares regression. We have changed the manuscript to make this clear and referenced our paper that we published in the American Journal of Public Health that similarly handled consumption scales from the California Healthy Kids Survey. However, we see similar results with hierarchical Poisson regression (see table below). We have added this information to the appendix.

Study 1 Fast-food Restaurant Patronage due to Nearness and Identification using Poisson hierarchical regression model

Predictor Variable

Relationship to Fast-food Restaurant Patronage

Fast-food restaurant near school

b = .08, df = 5980, z = 0.73, p = .47

Identification with student community

b = -.23, df = 5980, z = 2.94, p < .01

Near school × identification

b = .18, df = 5980, z = 2.25, p < .05

Reviewer comment: Every study should have limitations, but the text does not specify the limitations. To indicate which topics should be studied in the future, it would be necessary to identify the limitations of this paper.

 Author response: We have revised the Final Discussion section to include limitations. We added a related section on future research. We thank you for your comments.

Round 2

Reviewer 2 Report

I have no more futher comments and satisfied with this revised manuscript.